# Fungal Infections in the ICU during the COVID-19 Pandemic in Mexico

**DOI:** 10.3390/jof9050583

**Published:** 2023-05-18

**Authors:** Carla M. Roman-Montes, Saul Bojorges-Aguilar, Ever Arturo Corral-Herrera, Andrea Rangel-Cordero, Paulette Díaz-Lomelí, Axel Cervantes-Sanchez, Bernardo A. Martinez-Guerra, Sandra Rajme-López, Karla María Tamez-Torres, Rosa Areli Martínez-Gamboa, Maria Fernanda González-Lara, Alfredo Ponce-de-Leon, José Sifuentes-Osornio

**Affiliations:** 1Infectious Diseases Department, Instituto Nacional de Ciencias Médicas y Nutrición “Salvador Zubirán”, Mexico City 14080, Mexico; carla.romanm@incmnsz.mx (C.M.R.-M.); saul.bojorgs@gmail.com (S.B.-A.); wells_per@hotmail.com (E.A.C.-H.); bernardo.martinezg@incmnsz.mx (B.A.M.-G.); sandra.rajmel@incmnsz.mx (S.R.-L.); karla.tamezt@incmnsz.mx (K.M.T.-T.); 2Clinical Microbiology Laboratory, Instituto Nacional de Ciencias Médicas y Nutrición “Salvador Zubirán”, Mexico City 14080, Mexico; arco2069@gmail.com (A.R.-C.); paulettegdlomeli@gmail.com (P.D.-L.); axel.cervantess@incmnsz.mx (A.C.-S.); areli_martinez@hotmail.com (R.A.M.-G.); alf.poncedeleon@gmail.com (A.P.-d.-L.); 3General Direction, Instituto Nacional de Ciencias Médicas y Nutrición “Salvador Zubirán”, Mexico City 14080, Mexico; jose.sifuenteso@incmnsz.mx

**Keywords:** invasive fungal infection, COVID-19, ICU, CAPA, apergillosis, candidemia, SARS-CoV-2

## Abstract

**Background**: Invasive Fungal Infections (IFI) are emergent complications of COVID-19. In this study, we aim to describe the prevalence, related factors, and outcomes of IFI in critical COVID-19 patients. **Methods**: We conducted a nested case–control study of all COVID-19 patients in the intensive care unit (ICU) who developed any IFI and matched age and sex controls for comparison (1:1) to evaluate IFI-related factors. Descriptive and comparative analyses were made, and the risk factors for IFI were compared versus controls. **Results**: We found an overall IFI prevalence of 9.3% in COVID-19 patients in the ICU, 5.6% in COVID-19-associated pulmonary aspergillosis (CAPA), and 2.5% in invasive candidiasis (IC). IFI patients had higher SOFA scores, increased frequency of vasopressor use, myocardial injury, and more empirical antibiotic use. CAPA was classified as possible in 68% and 32% as probable by ECMM/ISHAM consensus criteria, and 57.5% of mortality was found. Candidemia was more frequent for *C. parapsilosis* Fluconazole resistant outbreak early in the pandemic, with a mortality of 28%. Factors related to IFI in multivariable analysis were SOFA score > 2 (aOR 5.1, 95% CI 1.5–16.8, *p* = 0.007) and empiric antibiotics for COVID-19 (aOR 30, 95% CI 10.2–87.6, *p* = <0.01). **Conclusions**: We found a 9.3% prevalence of IFIs in critically ill patients with COVID-19 in a single center in Mexico; factors related to IFI were associated with higher SOFA scores and empiric antibiotic use for COVID-19. CAPA is the most frequent type of IFI. We did not find a mortality difference.

## 1. Introduction

Invasive fungal infections (IFI) increase morbidity and mortality in COVID-19 patients even with timely diagnosis and treatment [1,2,3]. Patients with coronavirus disease (COVID-19) caused by the SARS-CoV-2 will become seriously ill and require entry to the intensive care unit (ICU). Older people and those with underlying medical conditions, such as cardiovascular disease, type 2 diabetes (T2DM), chronic respiratory disease, or cancer, are more likely to develop severe illnesses. The most frequent IFI described in patients with COVID-19 is COVID-19-associated pulmonary aspergillosis (CAPA) with a variable prevalence depending on the type of study; in observational studies, an average prevalence of 3.3% has been reported [4]. CAPA development does not require an immunocompromised host, and angioinvasion is not usually evident in contrast with other forms of invasive aspergillosis. CAPA diagnosis is a challenge because bronchoscopy with bronchoalveolar lavage (BAL) and galactomannan antigen (GM) testing and culture are usually warranted to improve the diagnostic yield. Additionally, bronchoscopies are rarely performed in COVID-19 patients due to the risk of disease transmission, the need for trained personnel, and the severity of critically ill patients in a prone position [5]. A systematic review described a CAPA mortality rate of 48.4% [6]. Several studies have shown increased rates of candidemia during the pandemic as many well-known risk factors, such as the use of corticosteroids, mechanical ventilation, broad-spectrum antibiotics, and central venous catheters, are highly prevalent among COVID-19 patients in the ICU [7,8,9,10]. As the COVID-19 pandemic progressed, invasive fungal infections have been increasingly described in critically ill patients, but despite the large number of publications that have studied their behavior, information in our region is limited [11].

Thus, we aim to describe the overall prevalence of IFI, CAPA, and candidemia, and the clinical characteristics, related factors, and outcomes of IFI in critical COVID-19 patients in the ICU.

## 2. Materials and Methods

We performed a nested case–control study between 27 March, 2020 to 21 September, 2021 in a tertiary center in Mexico City. We included adult patients (≥18 years old) with confirmed COVID-19 (symptoms plus compatible chest imaging CT-scan and positive SARS-CoV-2 real-time polymerase chain reaction (rt-PCR) result) admitted to the ICU. Patients who developed an IFI during their ICU stay were included as cases. Age- and sex-matched controls were randomly selected in a 1:1 ratio among ICU-admitted patients without IFI. Demographic, clinical, and microbiological data were obtained from the electronic medical records, and the cycle threshold (cT) values from the COVID-19 PCR test were provided by the local laboratory. 

CAPA is defined according to the ECMM/ISHAM (European Confederation of Medical Mycology (ECMM) and the International Society for Human and Animal Mycology (ISHAM) 2020 criteria. According to the ECMM/ISHAM category, probable CAPA requires a pulmonary infiltrate or nodules, preferably documented by chest CT, or cavitating infiltrate (not attributed to another cause), or both, combined with mycological evidence (mycologic criteria: ≥0.5 GM in serum or ≥1.0 GM in BAL or a positive BAL culture with *Aspergillus* species). Possible CAPA was considered with pulmonary infiltrate or nodules, preferably documented by chest CT, or cavitating infiltrate in combination with mycological evidence (e.g., microscopy, culture, or galactomannan, alone or in combination) obtained via non-bronchoscopy lavage (as tracheal aspirate). Both probable and possible should meet any of the following clinical findings: refractory fever for more than 3 days or a new fever after a period of defervescence of longer than 48 h during appropriate antibiotic therapy, in the absence of any other obvious cause; worsening respiratory status (e.g., tachypnoea or increasing oxygen requirements); hemoptysis; and pleural friction rub or chest pain, which can trigger diagnostic investigations for CAPA in patients with refractory respiratory failure for more than 5–14 days despite receiving all support recommended for patients with COVID-19 who are critically ill. [12]. We used standardized methods for the diagnosis as Platelia™ *Aspergillus* Ag and AsperGenius^®^ Species Multiplex real-time PCR kit. 

Invasive candidiasis (IC) is defined as the presence of any *Candida* sp. isolated in a sterile tissue (e.g., blood or sterile tissue). Pulmonary mucormycosis and cryptococcosis were considered according to the EORTC/MSGERC criteria (European Organization for Research and Treatment of Cancer/Mycoses Study Group Education and Research Consortium) [13]. We considered pharmacologic immunosuppression as a treatment for another comorbidity with any immunosuppressive medication, including corticosteroids. We recorded all the general and clinical characteristics as well as the episodes of COVID-19 and admission to the ICU.

Descriptive and comparative analyses were made. Baseline characteristics and risk factors were compared between groups (IFI vs. controls) using Chi-squared or Fisher’s exact test, and Student’s *t* or Mann–Whitney’s U test, according to their distribution. We described the odds ratio (OR) and 95% confidence interval (95% CI). Finally, a multivariate analysis was performed using a logistic regression model to identify factors associated with IFI. Variables with *p* < 0.2 in bivariate analysis were considered for the multivariable model. Missing data were not replaced. A *p* value < 0.05 was considered statistically significant. A post hoc sensitivity analysis for probable and possible CAPA cases was performed due to concerns regarding the low specificity of classification criteria. Statistical analysis was performed using Stata 12.0 Statistics software (College Station, TX, USA).

Due to the observational nature of the study, written informed consent was waived. The study was approved by the Institutional Review Board (Ref. Number 3691).

## 3. Results

From 27 March 2020 to 21 September 2021, 842 patients with COVID-19 were admitted to the ICU, of which 9.3% (78/842) developed an IFI during their ICU stay. In all patients, SARS-CoV-2 rt-PCR was positive, and all were under invasive mechanical ventilation (IMV). The most frequent IFI was CAPA in 60% (47/78) of the patients, followed by invasive candidiasis in 32% (25/78), pulmonary cryptococcosis in 5% (3/78), and pulmonary mucormycosis in 5% (3/78). 

Baseline demographic and clinical characteristics were similar among patients with IFI and controls (Table 1). Among patients with IFI and comorbidities, we found no differences between obese patients and those with hypertension. Pharmacologic immunosuppression was more frequent in IFI patients than in controls (9% (7/78) vs. 0%), *p* = 0.01); of the latter, three patients had connective tissue disorders, two had received a solid organ transplant (SOT), one had myasthenia gravis and one with chronic myeloid leukemia. Regarding other comorbidities, chronic obstructive pulmonary disease was present in 3% (5/78) vs. 1% (1/78) *p* = 0.56; chronic kidney disease (CKD) in 4% (3/78) vs. 0, *p* = 0.24; and chronic liver disease in 2.5% (2/78) vs. 0, *p* = 0.49; respectively.

Regarding COVID-19 index episodes, increased severity was more frequently reported in patients with IFI: APACHE II (Acute physiology and chronic health evaluation) score ≥ 10 (56% (44/78) vs. 54% (42/78), *p* = 0.74), SOFA (Sequential Organ Failure Assessment) score > 2 (92% (72/78) vs. 69% (54/80), *p* = <0.01), and use of vasopressor (90% (70/78) vs. 69% (54/78), *p* = 0.002). The use of dexamethasone as COVID-19 treatment was less frequent in patients with IFI (60% (47/78) vs. 72% (55/78), *p* = 0.12). COVID-19 treatment, ICU management, and outcomes are also described in Table 1. No differences in in-hospital mortality among patients with or without IFI were seen (46% (38/78) vs. 44% (34/78), *p* = 0.52). Factors related to IFI in a multivariable analysis (adjusted Odds ratio, aOR) were SOFA score > 2 (aOR 5.1, 95% CI 1.5–16.8, *p* = 0.007) and empiric antibiotic for COVID-19 (aOR 30, 95% CI 10.2–87.6, *p* = <0.01) (Table 1 and Table 2).

### 3.1. COVID-19-Associated Pulmonary Aspergillosis (CAPA)

CAPA was identified in 60% (47/78) of COVID-19 patients with IFI. The overall prevalence of CAPA was 5.6% (47/842). Following ECMM/ISHAM consensus criteria, 32% (15/47) were classified as probable CAPA and 68% (32/47) as possible. A detailed description of possible cases is provided in Appendix A, and a comparison between probable and possible cases is described in Table 3. Cases were suspected due to clinical worsening despite antibiotic treatment and ventilatory support; 30% (14/47) had persistent fever, 68% (32/47) recurrent fever, and 77% (36/47) presented worsening ventilatory parameters, despite antibiotic treatment. The radiologic criteria were met with chest CT in 66% (31/47) of cases; the most frequent findings were consolidations in 83% (26/31), ground glass opacities in 67% (21/31), nodules or micronodules in 19% (6/31), and cavitations in 3% (1/32); and the rest 34% (16/47) had lung infiltrates or radiographic worsening by chest ray, where CTs were unavailable due to hemodynamic or respiratory instability.

The mycologic criteria were fulfilled in all cases with different diagnostic methods: *Aspergillus* culture in 68% (32/47), BAL GM in 17% (8/47), serum GM in 13% (6/47), and positive *Aspergillus* RT-PCR in BAL in 2% (1/47). Regarding the positivity of GM, it is essential to modify the denominator; the test was not performed in all cases. Serum GM positivity was 23% (6/26) and BAL GM was 75% (6/8). Four patients with positive GM also had an *Aspergillus* culture. Of the total 38 *Aspergillus* isolates that were found (2 samples with two *Aspergillus* species in the same sample), 87% (33/38) were tracheal aspirates (TA) samples, 10.5% (4/38) from only BAL culture, and 3% (1/38) from BAL and TA. *Aspergillus fumigatus* from section *fumigati* was the most frequent species in 60.5% (23/38) followed by 16% (6/38) *Aspergillus* section *nigri*. Bacterial coinfections during ICU stay were diagnosed in 64% (30/47), of which 93% (28/30) had simultaneous bacterial ventilator-associated pneumonia (VAP) but only 30% (9/30) with bacterial isolate in a sample for fungal diagnosis. Antibacterial treatment was indicated in 83% (39/47) of patients with CAPA upon admission to ICU. Antifungal therapy was administrated in 85% (40/47), of which 50% (20/40) died, compared to 100% (7/7) with CAPA but without antifungal treatment who died (*p* = 0.01). The reason for not receiving treatment was a culture, or GM antigen results were available after death, and all were classified as possible CAPA. The most frequent antifungal was voriconazole in 85% (34/40), followed by amphotericin B in 7.5% (3/40), echinocandin in 5% (2/40) and isavuconazole in 2.5% (1/40). Overall mortality in CAPA was 57.5% (27/47) vs. 45% (34/78) in controls (*p* = 0.13). Possible CAPA cases were female more frequently, had increased disease severity according to APACHE II score, and received less frequent antifungal treatment. Possible cases had higher mortality compared to probable cases; however, this difference was not significant. Of the 20 patients who died, 35% (7/20) did not receive treatment vs. 65% (13/20) received antifungal treatment (*p* = 0.02); of the seven who did not receive treatment, 57% (4/7) were due to postmortem isolation of *Aspergillus* being reported (Table 3). 

### 3.2. COVID-19-Associated Candidiasis

Among 25 patients with invasive candidiasis, 88% (22/25) had candidemia, and 12% (3/25) had deep-seated candidiasis (two with empyema and one with abscessed pneumonia). There were a total of 27 *Candida* spp. isolates; among 22 candidemias, 9% (2/22) had mixed candidemia (more than 1 species was isolated). Of all Candida spp. isolates, 76% (19/25) were cultured from a central venous catheter (CVC), while the rest were cultured from non-CVC drawn blood; all CVC-related cases had a CVC removal within the first 48 h after the culture report. Fluconazole-resistant *Candida parapsilosis* was the most frequent species in 48% (13/27), followed by Candida albicans in 41% (11/27) and *Candida glabrata* in 11% (3/27). Regarding Candida risk factors, all had a central venous catheter and arterial line, 4% (1/25) had acute pancreatitis, and none received parenteral nutrition. Compared to controls, patients with IC underwent invasive procedures 16% (4/25) vs. 17% (13/78), *p* = 0.9, and received antibiotic treatment 100% (25/25) vs. 34% (27/78), [OR 35, CI 95% (4.5–273) *p* = 0.0001]) more frequently. Antifungal treatment was indicated in 96% (24/25), of which 87.5% (21/24) received echinocandins and 12.5% (3/24) received azoles. The median antifungal treatment duration was 16 days (IQR 13–18). Direct ophthalmoscopy was performed in 20% (5/25) of which two had abnormalities (vitritis and chorioretinitis). Transthoracic echocardiography was performed in 64% (16/25), and none met the endocarditis criteria. Overall, mortality was 28% (7/25) compared to 44% (34/78) in controls (*p* = 0.14). 

### 3.3. COVID-19-Associated Mucormycosis (CAM) and Cryptococcosis (CACr)

Three cases of pulmonary mucormycosis have been described. One case occurred in a patient with uncontrolled diabetes. Three cases of pulmonary cryptococcosis were diagnosed, of which none were associated with HIV or immunocompromised. Clinical characteristics are described in Table 4.

### 3.4. IFI Cases during the Time of Pandemic

As specific objectives, we describe in Figure 1 the incidence of IFI throughout the study period. CAPA incidence surveillance showed three outbreaks after June 2020. During each outbreak, suspended particles were measured, and the ventilation system was evaluated to identify possible sources of fungal spores. A lack of maintenance and change of filters in areas converted into ICUs was found. The filters of the reconverted areas were replaced in January 2021. After that, there was a decrease in the number of cases and the subsequent end of the converted units. Candidemia occurred as two outbreaks during which infection control practices and central line infection prevention bundles were reinforced, thereby achieving a decrease in the incidence.

## 4. Discussion

We found a prevalence of 9.3% of IFI in critically ill patients with COVID-19. CAPA and IC were the most common, while mucormycosis and cryptococcosis occurred infrequently. Our general IFI prevalence is within the range described in critical COVID-19 patients worldwide (7–26.6%) [14,15]. Distinct prevalence between geographic regions could be associated with study design, heterogeneity of patients, distinct local surveillance protocols, and the definitions used for classification. Gangneux J.P. et al. described a prevalence of 15% of IFIs; however, they performed a systematic twice-weekly screening for fungal infections including respiratory cultures and blood biomarkers, and patients were classified as colonized or proven/ probable CAPA as per the ECMM/ISHAM criteria, but found a lower prevalence (11%) when using AspICU criteria [14]. 

Additionally, one reason associated with underdiagnosis may be the lack of an established screening or epidemiological surveillance program. The justification for the variability has already been described by Kariyawasam R.M. et al., who, in their systematic review, analyzed the correlation between the CAPA definitions, which was low (correlation coefficient, ranging between 0.263 and 0.447) and even more critical than in the reviewed studies; 94 patients (33.9%) did not meet the criteria for any CAPA definitions. The low correlation reflects a discrepancy with the definitions used before the ECMM/ISHAM consensus criteria, on which many of the prevalence reports were based [16]. This finding highlights the importance of unifying the criteria. In our study, only the ECMM/ISHAM criteria were used, although all patients also met the AspICU criteria. Wide regional variations in CAPA have been described. The most relevant factor was the development of strict diagnostic criteria during the pandemic. Early reports described a prevalence of 25–30%; however, when cases were reclassified by the ECMM/ ISHAM 1–4% prevalence was found, which is closer to ours, as previously mentioned [16]. 

Nevertheless, the environment can also be an essential difference. As described by Matthias Egger et al., the possible factors associated with the variation in prevalence could be environmental exposure to *Aspergillus* and even a greater genetic predisposition, which may be different in different regions of the world. Other factors associated with low prevalence or underdiagnosis are the use of prophylaxis, the absence of screening, the low availability of studies such as bronchoscopy, and the lack of availability of studies such as the GM test [17]. In our center, a sudden increase in CAPA cases warranted inspection, a change of the ventilation duct filters, and an improvement in air quality conditions in reconverted ICUs, possibly leading to decreased case incidence [18]. 

The absence of screening, lack of widespread bronchoscopy availability, and non-availability of diagnostic biomarkers may result in an underestimated prevalence [19]. Most of our patients with CAPA were classified as possible, mostly based on tracheal aspirate samples; GM in BAL testing remains the test of choice. Although increasing certainty of the efficacy of protective equipment use and healthcare workers’ vaccination contributed to the increased availability of bronchoscopies, disease severity leading to hemodynamic instability or severe ARDS may still preclude obtaining BAL samples [17,20,21]. Of note, in this study, we found that possible CAPA cases received antifungal treatment less frequently than probable cases. Additionally, increased mortality, although non-significant, was found; although the “possible” definition has reduced diagnostic accuracy, sensitivity analysis of large cohorts may provide additional data on the impact of antifungal treatment on mortality. Kariyawasam RM et al. found increased survival in patients who received mold-active antifungals (survival rate 46.8% vs. 29.8%, *p* = 0.01); however, a meta-analysis of the association varied across the studies as reported by the authors, but the association did not reach statistical significance (OR 2.18; 95% CI 0.95–5.00, *p* = 0.6) and the studies were heterogeneous to reach conclusions, emphasizing that it is necessary to treat it when suspected, and an accurate diagnosis is necessary [16]. 

With these findings, it is not possible to draw conclusions; however, we could infer that whoever meets the CAPA criteria should receive antifungal treatment, with the hope that other studies can conclude this challenge by distinguishing which group benefits from the treatment. 

We found that patients who developed IFI were more severely ill as evidenced by higher SOFA, need for vasopressors, and prolonged IMV. In this study, pharmacologic immunosuppression, but no other comorbidities, were associated with IFI development in bivariate analysis. Other studies, such as the French MYCOVID study and a study in Germany by Leistner et al., also found an association between disease severity and IFI development [14,22]. Therefore, it could be concluded that the more severe the group of patients is, the greater the risk of developing IFI. Another factor that was found was the use of empirical antibiotics before admission to the ICU. Previous antibiotic use is associated with fungal infections such as candidiasis. Regarding other factors associated with IFIs, other factors favoring an increased prevalence are the widespread use of dexamethasone or tocilizumab. We did not find an association with corticosteroids for COVID-19 in initial reports after the RECOVERY trial [23,24]. However, recent multicenter prospective trials from France and Germany have shown that corticosteroid treatment is an independent risk factor for CAPA. Of note, recent trials use standardized CAPA definitions in contrast to the initial series [14,19]. Although steroid use was less frequent in patients who developed IFI, an association was not found in this study. More recently, Juergen Prattes et al., in a multicenter and multinational study, reported that CAPA was more prevalent among older patients receiving invasive ventilation and tocilizumab. CAPA is an independent and strong predictor of ICU mortality [25]. In our CAPA group, we found no difference regarding tocilizumab and mortality. 

Although patients with IFI did not have increased mortality, they had longer ICU and hospital stays, prolonged mechanical ventilation, and complications such as cardiac injury associated with COVID-19. One reason that there were no differences in mortality may be that we analyzed a set of patients with different IFIs whose mortality may not be comparable between each IFI; however, despite the trend towards higher mortality, we did not find statistical significance compared to the control group.

Specifically, in CAPA, mortality has been reported as high as 51.2%, as described in the meta-analysis by Shreya Singh et al. [26], or higher in the meta-analysis by Kariyawasam RM et al., where mortality in CAPA was 60% [16]. Some characteristics, especially of interest, are related to mortality in the subgroup of CAPA patients. A trend toward higher mortality in CAPA cases has been observed in various studies, including 61.8% (95% CI 50.0–72.8) in patients with CAPA versus 32.1% (27.7–36.7; *p* < 0.0001) without CAPA [14]. We found no difference in mortality.

Candidemia was the second most prevalent IFI. We found a prevalence that fits within the previously reported range in COVID-19 patients (2.5–5.1%) [27,28,29,30]. Several countries reported increased rates of candidemia during the COVID-19 pandemic, and in all cases, the ICU stay was a factor related to the need for ECMO support, although we did not show ECMO as a related factor because it was not used in our institution [9,31,32,33]. In Spain, an increased incidence of candidemia was seen during the first wave of the pandemic compared to other non-COVID-19 ICU patients. Typical risk factors for IC are widely prevalent among critically ill patients with COVID-19. Additionally, SARS-CoV2 infection alters innate immune cell activity, which may be predisposed to *Candida* infections [1]. Other reasons for the increased prevalence of candidemia during COVID-19 are Infection Control breaches and inappropriate use of personal protection equipment (PPE) [33]. After the first outbreak of candidemia, the Infection Control Department reinforced hand washing and catheter infection prevention bundles. After such measures, a dramatic decrease in the rate of IC was seen. Using venous catheters has been described as a risk factor in other studies with statistical significance, especially for femoral catheters. All our patients had a central venous catheter placed, and more than 70% had CVC-associated infection [34]. In our findings, invasive procedures were not different, but previous antibiotic use was more frequent in patients with candidemia compared to controls. Other studies did not find that the known factors for candidemia were different between the pre-COVID-19 era and the pandemic era; however, there was a tendency to have a more significant role between the use of steroids and previous colonization. It has been described that patient-patient transmission was infrequent due to the absence of clusters in its isolates. Moreover, what was evidenced was the lack of catheter care measures due to the high prevalence of catheter-related infection; this finding, what we believe, is due to an associated factor [35]. 

A case-level analysis by E. Seagle et al. found more frequent candidemia in patients with intensive care unit-level care, mechanical ventilation, and a central venous catheter; receipt of corticosteroids and immunosuppressants were each >1.3 times more common in patients with COVID-19; 100% of our patients had all three factors in this case [36]. In Florida, colonization was detected in 17% of ICU patients, and several Infection Control breaches associated with PPE overuse and lapses in hand hygiene and disinfection were noticed, but as a disadvantage, we did not investigate previous colonization [37,38]. 

Additionally, outbreaks of *Candida auris* were detected during the pandemic in several countries (Italy, Lebanon, USA, China, Brazil, Colombia, and Spain). In a systematic review, they found that patients with *C. auris* were more diabetic, hypertense, and obese; furthermore, the traditional factors of candidemia were found in patients with *C. auris*, such as having a central venous catheter (76.8%), intensive care unit (ICU) stay (75.6%), and broad-spectrum antibiotic usage (74.3%). These findings highlight the importance of these factors no matter the species of *Candida* involved [38,39,40,41]. Additionally, the first report of *C. auris* in Mexico was described during the pandemic, causing an outbreak; fortunately, in our center, we did not identify cases due to *C. auris* [42]. 

Regarding IC mortality, hypothesized that one possibility is the low mortality among patients with candidemia, probably due to the predominance of *C. parapsilosis*. This finding is similar to that of Horn et al., who found that *C. parapsilosis* was one of the non-albicans species with the lowest mortality (23%, *p* = 0.001) and was also associated with the fact that *C. parapsilosis* less frequently affected hematological or highly immunosuppressed patients, considering that it was not precisely our affected population [43,44]. The other mortality risk factor is early CVC removal; in our patients, it was withdrawn or changed shortly after the diagnosis. Catheter removal has been associated with lower mortality in patients with candidemia [45]. 

In the literature, the increased mortality among COVID-19 patients with candidemia was related to a low incidence of timely antifungal treatment, increased age, sepsis, and previous corticosteroid treatment, and we only found a relation with dexamethasone for COVID-19 treatment in IC patients [7]. Of note, the most frequent isolate was a Fluconazole-resistant *C. parapsilosis* strain, identified for the first time in our institution. Although it was not possible to search for the resistance mechanism, this clone was described in a nearby hospital in Mexico City in 2020, which could represent sustained colonization and transmission among local healthcare institutions [46]. The emergence of fluconazole-resistant *C. parapsilosis* has been previously described [47,48].

We found a very low prevalence of COVID-19-associated mucormycosis (CAM), in fact only in pulmonary cases, despite being a country with a high prevalence of type 2 diabetes mellitus. In contrast, in India, a country with a high prevalence of rhino-orbito-cerebral mucormycosis, due in part to a high prevalence of type 2 diabetes mellitus, the occurrence of multiple cases in patients with COVID-19 has led to an international alert by the WHO (World Health Organization) [49]. A prevalence of 12% was reported in a single-center ICU. Rhino-orbito-cerebral CAM should be described as associated with COVID-19 patients, but in this study, only pulmonary cases were detected, which represent less than 10% of CAM reported cases. Cases have been reported among immunocompromised hosts with worsening respiratory insufficiency, progressive pneumonia, persistent fever, and organ failure [50].

We described three cases of pulmonary cryptococcosis, which also had meningeal disease. Two of the patients died. Although they were reported as previously healthy, they were all older adults. The association between cryptococcosis and COVID-19 remains anecdotic. Thirteen cases were described in a recent review, of which only one had HIV (CD4+ cell count not reported) and 92% received steroids for severe COVID-19. Mortality was 53.8%. Pulmonary, meningeal or both presentations have been described. Severe COVID-19 with mechanical ventilation and immunosuppressors may have a role in cryptococcal reactivation, but further research is required [51].

We recognize our limitations, the retrospective nature of the study, and the possibility of missing data. However, it is a single-center study, which decreases its external validity for patients with severe COVID-19 that may be similar in other centers. We recognize as a limitation including the possible CAPA cases, but comparing the outcomes and characteristics with probable CAPA cases, they are similar; however, we may have a misclassification bias; we consider that including possible and probable CAPA are more epidemiologically representative. Possible strengths are the comparison with the group without IFI, the extensive review of compliance with the criteria in CAPA cases, and obtaining *Candida* and *Aspergillus* cultures from a reference microbiology laboratory.

It is likely various environmental and host factors are involved in IFI development during severe COVID-19. Increased surveillance of ventilation filters should be made in reconverted ICUs. The impact on the number of CAPA cases after improving such ventilation was out of the scope of this study. Additionally, the ongoing construction of a novel building within our hospital may have impacted the increased incidence [52]. While CAPA remains the dominant IFI, individual factors may contribute to specific IFIs, such as uncontrolled T2DM for CAM or cellular immunosuppression for cryptococcosis. High mortality, as well as increased ICU and hospital stay, prolonged mechanical ventilation, and increased healthcare costs represent the increased burden of IFIs. Improved diagnostic algorithms and early antifungal treatment are needed to reduce mortality [53].

## 5. Conclusions

In our center, we found a 9.3% prevalence of IFIs in critically ill patients with COVID-19 during first waves of pandemic; factors related to IFI were associated with higher SOFA scores and empiric antibiotics for severe COVID-19. CAPA was the most frequent type of IFI. Invasive candidiasis was presented as an outbreak. Mortality was not statistically or significantly higher among patients with IFI. Infection control breaches should be reinforced during pandemics to reduce the prevalence of IFI.

## Figures and Tables

**Figure 1 jof-09-00583-f001:**
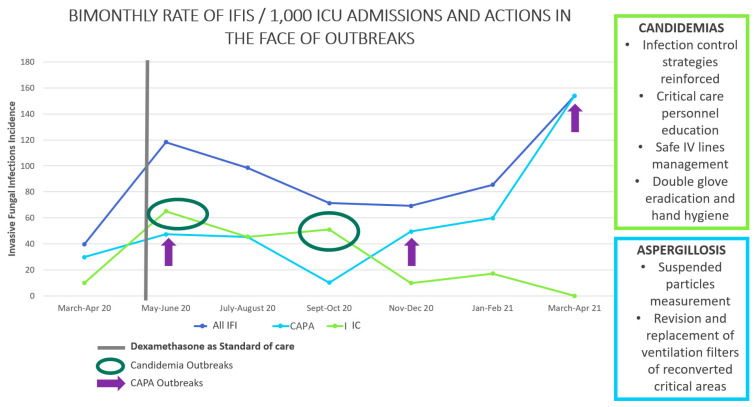
Invasive fungal infections in ICU rates and local interventions. Figure 1 describes the rate of cases of bimonthly IFIs per 1000 ICU admissions. We specified the moment of initiation of dexamethasone as standard care in the first wave of the pandemic, the moments of increase in cases of candidiasis, and three moments of increase in cases of aspergillosis. We show in the same figure the actions that were carried out in support of hospital epidemiology when these cases were identified. Aspergillosis prevention measures were performed until 2021.

**Table 1 jof-09-00583-t001:** Clinical characteristics and related factors with IFI of critical COVID-19 patients.

Characteristics	General *n* = 156 (%)	IFI *n* = 78 (%)	Controls *n* = 78 (%)	Bivariate *p*
Age, mean (SD)	56.6 (14.1)	56.5 (14.2)	57.8 (14.1)	0.9
Male Sex	112 (72)	56 (72)	56 (72)	1.0
Comorbidities				
Obesity	81 (52)	42 (54)	39 (50)	0.63
Overweight	56 (36)	28 (36)	28 (36)	0.95
Arterial hypertension	53 (34)	27 (35)	26 (33)	0.86
Type 2 diabetes mellitus	49 (32)	23 (29.5)	26 (38)	0.56
Immunosuppression	7 (4.5)	7 (9)	0	**0.01**
Pulmonary disease	7 (4.5)	5 (6)	2 (2.5)	0.44
Smoking	23 (15)	13 (17)	10 (13)	0.49
Charlson ≥ 4	22 (14)	12 (15)	10 (13)	0.65
Severity scores				
APACHE II Score ≥ 10	73 (47)	38 (49)	35 (45)	0.63
SOFA score > 2	126 (81)	72 (92)	54 (69)	<0.01
Lymphopenia *	107 (69)	49 (63)	58 (74)	0.12
SARS-CoV-2 PCR Ct < 24	52 (33)	27 (35)	25 (32)	0.73
COVID-19 treatment				
Tocilizumab	10 (27)	7 (37)	3 (17)	0.16
Remdesivir	4 (11)	2 (10.5)	2 (10.5)	0.95
Empiric antibiotic	99 (63.5)	72 (92)	27 (35)	**<0.01**
Dexamethasone	103 (66)	47 (60)	56 (72)	0.12
Hospital stay, mdn (IQR)	26 (17–41.5)	34.5 (23–51)	21 (12–29)	**<0.01**
ICU stay, mdn (IQR)	18 (11–28)	22 (16–41)	13 (7–24)	**<0.01**
IMV duration, mdn (IQR)	15.5 (9.5–25.5)	20 (14–31)	11 (7–22)	**<0.01**
Prolonged IMV (>21 days)	56 (36)	35 (45)	21 (27)	**0.02**
Prone position	115 (74)	61 (78)	54 (69)	0.20
Vasopressors	124 (79.5)	70 (90)	54 (69)	**0.002**
Hemodialysis	12 (8)	9 (11.5)	3 (4)	0.07
Myocardial injury	24 (15)	17 (22)	7 (9)	**0.03**
Pulmonary Venous thrombosis	21 (13.5)	12 (15)	9 (11.5)	0.48
Bacterial co-infection ^$^	92 (59)	49 (63)	43 (55)	0.33
VAP	77 (87.5)	38 (84)	39 (91)	0.37
In-hospital mortality	72 (46)	38 (49)	34 (44)	0.52

IFI: invasive fungal infection, ICU: intensive care unit, IQR: interquartile range, mdn: median, IMV: invasive mechanical ventilation, SD: standard deviation, VAP: ventilator-associated pneumonia. * Lymphopenia was defined as Lymphocytes <1000 × 10^3^/μL. $: From patients with bacterial coinfection in ICU, only 9 with bacterial isolation in the same clinical sample from fungal identification.

**Table 2 jof-09-00583-t002:** Factors associated with invasive fungal infections in critical COVID-19 patients in a multivariate analysis.

Variables	Multivariate Analysis
**SOFA score > 2**	**aOR 5.1, 95% CI 1.5–16.8, *p* = 0.007**
**Empiric antibiotic**	**aOR 30, 95% CI 10.2–87.6, *p* = <0.01**
**Prolonged IVM**	aOR 2.3, 95% 0.9–6.2, *p* = 0.09
**Dexamethasone**	aOR 0.81, 95% CI 0.31–2.1, *p* = 0.68
**Lymphopenia**	aOR 0.56, 95% CI 0.20–1.5, *p* = 0.27

aOR: adjusted Odds Ratio, CI: confidence intervals; IVM; invasive mechanical ventilation. Note: Pseudo R: 0.4, Number of observations: 156.

**Table 3 jof-09-00583-t003:** Clinical characteristics between probable and possible COVID-19-associated pulmonary aspergillosis (CAPA) cases.

Characteristics	All CAPA Cases *n* = 47 (%)	Probable CAPA *n* = 15 (%)	Possible CAPA *n* = 32 (%)	*p* Bivariate
Male sex	34 (72)	14 (93)	20 (62.5)	**0.02**
Age, mean (SD)	57.3 (14.3)	53.3 (15.9)	59.2 (13.4)	0.18
Obesity	23 (49)	6 (40)	17 (53)	0.40
Immunosuppression	6 (13)	3 (20)	3 (9)	0.30
Severity scores				
APACHE II Score ≥ 10	23 (49)	3 (20)	20 (62.5)	**0.007**
SOFA score > 2	45 (96)	15 (100)	30 (94)	0.32
Lymphopenia	31 (66)	8 (53)	23 (72)	0.21
Empiric antibiotic previous ICU	42 (89)	14 (83)	28 (87.5)	0.54
Dexamethasone	30 (64)	10 (67)	20 (62.5)	0.78
CAPA diagnosis				
Persistent Fever	14 (30)	7 (47)	7 (22)	0.08
Recurrent Fever	32 (68)	9 (60)	23 (72)	0.41
Worsening ventilatory parameters	36 (77)	12 (80)	24 (75)	0.70
Chest CT for diagnosis	31 (66)	11 (73)	20 (62.5)	0.46
Antifungal treatment	40 (85)	15 (100)	25 (78)	0.05
Death	27 (57.5)	7 (47)	20 (62.5)	0.30

CAPA: COVID-19-associated pulmonary aspergillosis; CT: computed tomography, ICU: Intensive care unit, SD: Standard deviation, VAP: Ventilator-associated pneumonia. * Lymphopenia was defined as Lymphocytes <1000 × 10^3^/μL.

**Table 4 jof-09-00583-t004:** General and clinical characteristics of COVID-19-associated mucormycosis (CAM) and cryptococcosis (CACr).

Characteristic/Patient	CAM Patient 1	CAM Patient 2	CAM Patient 3	CACr Patient 1	CACr Patient 2	CACr Patient 3
Sex	Male	Male	Male	Male	Male	Male
Age	62	68	51	51	86	70
Comorbidity	Obesity	Talcum pulmonary silicosis	Uncontrolled Diabetes	Uncontrolled Diabetes	Health previously	Health previously
Risk Factor	None	Lymphopenia	Lymphopenia	Lymphopenia	None	None
Dexamethasone for COVID-19	Yes	Yes	Yes	Yes	Yes	Yes
Site of infection	Pulmonary/ Sinusitis	Pulmonary	Pulmonary/ Sinusitis	Pulmonary/ meningeal	Pulmonary	Pulmonary/ Pleural
Clinic	Fever, anosmia, dysgeusia, cyanosis without dyspnea	Dry cough, sore throat, fever, dyspnea	Cephalea, dry cough, arthralgia, myalgia, dyspnea	Fever (38.5 °C), dry cough, dyspnea	Dyspnea, backache, cephalea	General discomfort, diarrhea, dyspnea
CT finding	Thorax: progression of pneumonia, crazy paving, consolidation Paranasal: Pansinusitis	Thorax: progression of pneumonia, micronodular Paranasal: none	Thorax: progression of pneumonia, ground glass opacities, consolidation CT paranasal: Pansinusitis	Thorax: progression of pneumonia, ground glass opacities Brain MRI: bilateral frontotemporal pachymeningitis	Thorax: progression of pneumonia, ground glass opacities, and apical nodule	Thorax: pulmonary embolism, ground glass opacities, subpleural cyst, pneumatocele, hydro-pneumothorax
Mycologic	*Rhizopus* spp. in TA	*Rhizopus oryzae* in TA	*Lichteimia corymbifera TA*	*Cryptococcus neoformans* in TA Negative antigen LF in CSF	*Cryptococcus neoformans* in TA	*Cryptococcus neoformans* in pleural effusion
Treatment	Posaconazole	Isavuconazole	Isavuconazole	Amphotericin B + Fluconazole	Non-treatment (post-mortem diagnosis)	Fluconazole
Outcome	Dead	Alive	Dead	Alive	Dead	Dead

CT: computed tomography, CSF: cerebrospinal fluid, LF: lateral flow, MRI: magnetic resonance imaging; TA: tracheal aspirate. * Lymphopenia was defined as Lymphocytes <1000 × 10^3^/μL.

## Data Availability

Data supporting reported results can be provided by the corresponding authors upon request.

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
