# Peer review of "Fungal Infections in the ICU during the COVID-19 Pandemic in Mexico"

_jof, 2023, doi:10.3390/jof9050583_

Round 1
Reviewer 1 Report
Roman-Montes and colleagues submit a manuscript entitled “Fungal Infections in the ICU during the COVID-19 pandemic in Mexico”. The manuscript is reasonably well written.
Comments:
- In the abstract, please define abbreviations that are not obvious, such as CAPA, Fluco, R, IMV, and IC.
- Abbreviations will need to be re-defined in the body of the manuscript of course, as you have done. However, within the body of the manuscript, they only need to be defined once. For example, CAPA is defined on line 34 and then again on line 63 and line 112.
- Line 32: define Covid-19.
- Line 46: “As the Covid-19 pandemic progresses…”. WHO has declared the pandemic over, although it remains a global emergency. Perhaps the word progresses should be changed to past tense, progressed.
- Line 46: define ICU.
- Line 57: define SARS-CoV-2
- Line 64: define ECMM/ISHAM
- Line 65: define GM
- Line 71: you do not need to capitalize Isolate
- Line 72: you do not need to capitalize Mucormycosis or Cryptococcosis
- Line 73: define EORTC/MSGERC
- Line 86: delete extra blank paragraph
- Line 92: you do not need to capitalize Invasive Candidiasis
- Line 99: you do not need to capitalize Obstructive Pulmonary Disease
- Line 103: define APACHE
- Line 103: define SOFA
- Line 109: define aOR
- Lines 110-111: you do not need to capitalize Invasive Mechanical Ventilation
- Line 114: change follow to following
- Line 118: define CT
- Line 124: define RT-PCR
- Lines 126-128: place “Aspergillus”, “fumigatus”, “fumigati”, and “nigri” in italics
- Table 2: clarify in the title if this is a univariate or multivariate analysis.
- Table 3: “spp.” does not need to be in italics.
- Line 232: define “AspICU”
- Line 252: Galactomannan does not need to be capitalized
- Line 368: define WHO
- Line 390: define T2DM

Author Response
Dear Reviewer 1 and editor,
We appreciate the time in review and valuable comments; here, you will find our answers and changes.
Comments:
- In the abstract, please define abbreviations that are not obvious, such as CAPA, Fluco, R, IMV, and IC.
R: We already define abbreviations in the abstract. (Lines 16,19,20).
- Abbreviations will need to be re-defined in the body of the manuscript of course, as you have done. However, within the body of the manuscript, they only need to be defined once. For example, CAPA is defined on line 34 and then again on line 63 and line 112.
R: We removed repeated abbreviation definitions, thanks. (Line 63 and 112)
- Line 32: define Covid-19.
R: We added a definition of COVID-19 and complications of illness. (Line 34-37)
- Line 46: “As the Covid-19 pandemic progresses…”. WHO has declared the pandemic over, although it remains a global emergency. Perhaps the word progresses should be changed to past tense, progressed.
R: We make the change (Line 51)
- Line 46: define ICU.
R: It was already defined (Line 37) ” entry to intensive care unit (ICU)”.
- Line 57: define SARS-CoV-2
R: It was already defined (Line 35)
- Line 64: define ECMM/ISHAM
R: It was already defined (Line 69-70) “ECMM/ISHAM (European Confederation of Medical Mycology (ECMM) and the International Society for Human and Animal Mycology (ISHAM) ) 2020 criteria.”
- Line 65: define GM
R: It was already defined (Line 72)
- Line 71: you do not need to capitalize Isolate
R: We make the change (Line 79)
- Line 72: you do not need to capitalize Mucormycosis or Cryptococcosis
R: We make the changes (Line 80)
- Line 73: define EORTC/MSGERC
R: It was already defined (Line 72) “EORTC/MSGERC criteria (European Organization for Research and Treatment of Cancer/Mycoses Study Group Education & Research Consortium)”
- Line 86: delete extra blank paragraph
- Line 92: you do not need to capitalize Invasive Candidiasis
R: We make the changes (Line 101)
- Line 99: you do not need to capitalize Obstructive Pulmonary Disease
R: We make the change (Line 108)
- Line 103: define APACHE, “APACHE II (Acute physiology and chronic health evaluation”
R: It was already defined (Line 112)
- Line 103: define SOFA, “SOFA (Sequential Organ Failure Assessment) score”
R: It was already defined (Line 113)
- Line 109: define aOR
R: It was already defined (Line 119) “(adjusted Odds ratio, aOR)”
- Lines 110-111: you do not need to capitalize Invasive Mechanical Ventilation
R: We make the change (Line 121)
- Line 114: change follow to following
R: We make the change (Line 125)
- Line 118: define CT, “computed tomography (CT)”
R: It was already defined (Line 75)
- Line 124: define RT-PCR
R: It was already defined (Line 67), “polymerase chain reaction (rt-PCR)”
- Lines 126-128: place “Aspergillus”, “fumigatus”, “fumigati”, and “nigri” in italics
R: We make the change (Line 136-140)
- Table 2: clarify in the title if this is a univariate or multivariate analysis.
R: We made the change in the title. “Table 2. Factors associated with Invasive Fungal Infections in critical COVID-19 patients in a multivariate analysis.”
- Table 3: “spp.” does not need to be in italics.
R: We make the change (Table 3)
- Line 232: define “AspICU”
- Line 252: Galactomannan does not need to be capitalized
R: We make the change
- Line 368: define WHO
R: It was already defined (Line 359), “(World Health Organization)”
- Line 390: define T2DM
R: It was already defined (Line 38), “type 2 diabetes”

Reviewer 2 Report
This is an interesting paper. It is always good to see data from other countries to look at the variance, more globally.
The biggest issue with this paper is the classification of cases and controls. The majority of CAPA cases are possible. That means they could be anything else (another pathogen, organising pneumonia post COVID). This creates misclassification bias. The implications of this are that the down stream findings (results) are affected by the misclassification and the findings may not be accurate. The best way to fix this is to leave these cases out. Bacterial co-infection was detected in 62.5% so these infections may overlap. The cases may not be CAPA but could be bacterial infection. Bacteria can cause similar CT scan findings to COVID and CAPA.
In addition, you didn’t state the definition of possible CAPA. As far as I can see you only included the probable definition and what CAPA is not.
The radiological features were met in only 64.5% of cases. If you are relying on that for possible cases then how did you make the diagnosis of possible cases otherwise?This may further increase the misclassification bias.
Abstract; the last sentence talks about infection control breaches; yet, there is nothing else in the results section of the abstract about this. If it is not included in the results, you can't have it as a conclusion. The sentence kind of hangs there too. it is not a conclusion; it is additional information. This sentence needs to be changed. I would delete it as it is distracting/confusing. But if you wish to retain it, then put results in the results section and say something like good infection control practices need to be maintained to prevent candidaemia. That is a conclusion.
Results
How the SARS-CoV-2 was diagnosed needs to be clear. Did all get a SARS-CoV-2 positive PCR result? If PCR was not used to make a diagnosis then how can we be sure that all cases are COVID and not other infections, like RSV or influenza? Please make this crystal clear.
It is very interesting that less dexamethasone was used in IFI cases as compared with controls. This seems contrary to other studies. I can’t see that this is discussed in the discussion section. It needs to be as it is a very significant finding. Need to provide a hypothesis for this. Could it be due to the classification bias that I pointed out earlier?
You report that GM was positive in 6% of cases but your denominator is the total group (n=48). This should really be reported as a percentage of the bronchoscopies performed. This is because not many BALs were performed during COVID. This denominator will give us much more information around the utility of BAL in diagnosing CAPA. Same goes for Aspergillus RT-PCR. Need to state what PCR methodology you used as well. A reference will be fine. Need to see if it is compliant with the EAPCRI method or not. If it is not an EAPCRI compliant methodology then this may explain the difference in rate of positivity compared with BALGM.
You did use serum GM so could have used this to diagnose probable cases, especially since BALs were not used.
Was a CT scan repeated at the time of diagnosis of CAPA? This is to see changes form baseline.
Lines 95 and 96 – it is unclear what you are trying to say. Are you comparing IFI cases to the overall cohort or to the controls. Table 1 looks like you are comparing to controls. Please modify the sentence so we know what is being compared.
Line 79 – should this be in the abstract?
Can you include in the discussion as to why you had very high cases of CAPA at the end of the study period when you had put in place interventions to control outbreaks? Could it be that the filters needed changing again? Could it be another factor. Need to discuss in the discussion section.
What do you mean by excessive PPE? Can PPE ever be excessive? Do you mean improved PPE use?
General corrections:
In several places you have capitalised words that don’t need to be. For example – line 99 Chronic Obstructive Pulmonary Disease – should be
Chronic obstructive pulmonary disease. Same in lines 92, 111.
Also all fungal names have a nomenclature, which means that they should be in italics. Please ensure that they are all in italics throughout the paper.
Line 339 – the word should be hypertensive not hypertensed.
Author Response
Dear Reviewer 2 and editor,
We appreciate the time in review and valuable comments; here, you will find our answers and changes.
The biggest issue with this paper is the classification of cases and controls. The majority of CAPA cases are possible. That means they could be anything else (another pathogen, organizing pneumonia post COVID). This creates misclassification bias. The implications of this are that the downstream findings (results) are affected by the misclassification and the findings may not be accurate. The best way to fix this is to leave these cases out. Bacterial co-infection was detected in 62.5% so these infections may overlap. The cases may not be CAPA but could be bacterial infection. Bacteria can cause similar CT scan findings to COVID and CAPA.
R: We agree that possible cases have lower diagnostic certainty and may represent other etiology. However de decided to include this category to reflect the widespread limitation of bronchoscopy availability. Also, the ECMM/ISHAM classification includes this category and even recommends antifungal therapy. Of note, among cases with bacterial coinfection, the inclusion criteria considers those who are not improving despite appropriate therapy. Possible cases may reflect the most frequent scenario during the COVID-19 pandemic.
In addition, you didn’t state the definition of possible CAPA. As far as I can see you only included the probable definition and what CAPA is not.
R: We added the possible CAPA definition, “was considered with pulmonary infiltrate or nodules, preferably documented by chest CT, or cavitating infiltrate in combination with mycological evidence (eg, microscopy, culture, or galactomannan, alone or in combination) obtained via non-bronchoscopic lavage (as tracheal aspirate)”. (Lines 75-78).
The radiological features were met in only 64.5% of cases. If you are relying on that for possible cases, then how did you make the diagnosis of possible cases otherwise? This may further increase the misclassification bias.
R: All the patients met the radiological criteria, in 64.5% with CT and we specified that “35.5% (17/48) met the radiologic criteria with chest X ray, with alterations or radiographic worsening by chest ray, where CTs were unavailable due to hemodynamic or respiratory instability”. Considering the ECMM/ISHAM recommendations that say preferably documented by chest CT; the guidelines do not mention that not having TC is an exclusion criterion if it meets the other criteria. Further 6 of 17 cases that only have X ray, have a positive serum GM.
Abstract; the last sentence talks about infection control breaches; yet, there is nothing else in the results section of the abstract about this. If it is not included in the results, you can't have it as a conclusion. The sentence kind of hangs there too. it is not a conclusion; it is additional information. This sentence needs to be changed. I would delete it as it is distracting/confusing. But if you wish to retain it, then put results in the results section and say something like good infection control practices need to be maintained to prevent candidaemia. That is a conclusion.
R: We agree this sentence is not an appropriate conclusion. We removed it from the abstract and refer to this likely explanation in the discussion section.
Results
How the SARS-CoV-2 was diagnosed needs to be clear. Did all get a SARS-CoV-2 positive PCR result? If PCR was not used to make a diagnosis then how can we be sure that all cases are COVID and not other infections, like RSV or influenza? Please make this crystal clear.
R: We modified the results section to specify that 99% of the cases had positive SARS-CoV-2 PCR and one SARS-CoV-2 PCR negative case had a negative respiratory panel (RespiFinder® SMART 22). (Line 101-102)
It is very interesting that less dexamethasone was used in IFI cases as compared with controls. This seems contrary to other studies. I can’t see that this is discussed in the discussion section. It needs to be as it is a very significant finding. Need to provide a hypothesis for this. Could it be due to the classification bias that I pointed out earlier?
R: We agree it is an unlikely finding. An independent association was not found between steroid use and IFI, it is likely that the finding in univariate analysis is due to confussors. We modified the discussion section as follows “Although dexamethasone use was less frequent in cases in univariate analysis. An independent association was not found in multivariate analysis.
You report that GM was positive in 6% of cases but your denominator is the total group (n=48). This should really be reported as a percentage of the bronchoscopies performed. This is because not many BALs were performed during COVID. This denominator will give us much more information around the utility of BAL in diagnosing CAPA. Same goes for Aspergillus RT-PCR. Need to state what PCR methodology you used as well. A reference will be fine. Need to see if it is compliant with the EAPCRI method or not. If it is not an EAPCRI compliant methodology then this may explain the difference in rate of positivity compared with BALGM. You did use serum GM so could have used this to diagnose probable cases, especially since BALs were not used.
R: Right; the purpose was to explain that this percentage is the proportion of cases diagnosed by this method and not the positivity of GM. The text clarifies that the first percentages were about how to diagnose cases and we add the positivity of GM explanation. “Regarding the positivity of GM, it is essential to modify the denominator; not in all cases performed the test, serum GM positivity was 38.5% (5/13) and BAL GM 67% (6/8)”. (Lines 145-146).
Only one case was diagnosed by Aspergillus RT-PCR; we used to follow the provider recommendations of the AsperGenius® Species Multiplex real-time PCR kit.
We specify in line 144 that the mycological criteria were met with GM serum 10% (5/48)
Was a CT scan repeated at the time of diagnosis of CAPA? This is to see changes form baseline.
R: CT scan was repeated in 31 cases, CTs were at the time to diagnosis CAPA with the alterations mentioned and not presented previously at time to COVID-19 diagnosis.
Lines 95 and 96 – it is unclear what you are trying to say. Are you comparing IFI cases to the overall cohort or to the controls. Table 1 looks like you are comparing to controls. Please modify the sentence so we know what is being compared.
R: We clarified that it was compared between those who developed IFI vs. controls (without IFI).
Line 79 – should this be in the abstract?
R: We add to the abstract that part of the methods.
Can you include in the discussion as to why you had very high cases of CAPA at the end of the study period when you had put in place interventions to control outbreaks? Could it be that the filters needed changing again? Could it be another factor. Need to discuss in the discussion section.
R: We included the following statement in the discussion section: Increased surveillance of ventilation filters should be made in reconverted ICUs. The impact on the number of CAPA cases after improving such ventilation was out of the scope of this study. Also, ongoing construction of a novel building withing our hospital may have impacted on increased incidence. Lines 483-486
We had more cases at the end of the study, close to the date of the change of filters and prevention measures. Our limitation is that the impact of the measures should be reported prospectively in this work, although the following case of CAPA not included in this work was six months after the actions. Lines 391-394.
What do you mean by excessive PPE? Can PPE ever be excessive? Do you mean improved PPE use?
R: At the beginning of the pandemic, in many countries, excessive PPE such as double gloving or gowns leading to suboptimal hand hygiene was seen and reported by some authors. We modified the word excessive to inappropriate for a better understanding. Also, we included a reference from a candidemia outbreak in Miami related to inappropriate PPE use. (Prestel C. et al. MMWR Morb Mortal Wkly Rep 2021;70:56–57)
General corrections:
In several places you have capitalised words that don’t need to be. For example – line 99 Chronic Obstructive Pulmonary Disease – should be Chronic obstructive pulmonary disease. Same in lines 92, 111.
R: We corrected incorrect capitalized words.
Also all fungal names have a nomenclature, which means that they should be in italics. Please ensure that they are all in italics throughout the paper.
R: We corrected italics for fungal names.
Line 339 – the word should be hypertensive not hypertensed.
R: We made the correction.

Round 2
Reviewer 2 Report
Thanks for the corrections.
I understand about the difficulties in getting CT scans and BAL examinations. However, we must still correctly assign a category to cases and be certain (or as certain as we can be) of the accuracy.
I remain concerned about the possible cases. Not all cases had a CT and only 6/17 (35%) had radiological signs consistent with IFD. How many of these possible cases had positive GM, and/or microscopy and/or culture? Need to tabulate; so, we can see how solid are these possible cases.
Deterioration despite appropriate antibiotics is not an adequate diagnostic criterion for IFD. This was used in empiric antifungal therapy studies and was well proven to be inaccurate. I think this is not an adequate criterion to diagnose IFD. Failure to respond to antibiotics could be due to anything. It is too non-specific.
I still think the cases of possible aspergillosis need to be removed. There are valid and good reasons for this as I outlined here and in my previous review.
However, one way to solve any potential obstacle/impasse to this is to do a sensitivity analysis. For this you compare outcomes for possible cases and probable/proven cases, if there is a discrepancy (possible cases have a greater survival) then the misclassification bias is present. The results need to be shown of this sensitivity analysis in the paper. That way the reader can make up their mind about the value of these cases and the solidity of the data.
There was one case that was SARS-CoV-2 and yet was included. This case need to be removed. yes, otehr virsues were negative too but that does not mean they had SARS-CoV-2. This does not fit the criteria for COVID- assoicated IFD. This cases needs to be removed and the data reanalysed.
Please remove the word excessive before PPE. There is no such thing. What you are trying to say is that the PPE measures may have contributed to healthcare associated candidaemia.
